# Work alienation influences nurses' readiness for professional development and willingness to learn: A cross-sectional correlation study

Othman A. Alfuqaha[1]*, Ohood F. Shunnar[2], Reema A. Khalil[3], Fadwa N. Alhalaiqa[4], Yazan Al Thaher[5], Uday M. Al-masarwah[1], Tareq Z. Al Amad[5]

**1** Counseling and Mental Health Department, Faculty of Educational Sciences, The World Islamic Sciences & Education University, Amman, Jordan, **2** Department of Nursing, Jordan University Hospital, The University of Jordan, Amman, Jordan, **3** Princes Basma Comprehensive Health Care Center, Jordanian Nurses and Midwifery Council, Amman, Jordan, **4** Faculty of Nursing, Philadelphia University, Amman, Jordan, **5** Oral Maxillo-Facial Surgery Department, Jordan University Hospital, The University of Jordan, Amman, Jordan

☯ These authors contributed equally to this work.

\* Othman.fuqaha@wise.edu.jo

## Abstract

Work alienation has a negative impact on nursing profession and may affect professional nursing development and willingness to learn during the era of coronavirus disease 2019 (COVID-19). The aim of this study was to explore the perceived levels of professional development, willingness to learn, and work alienation during this pandemic among nurses in Jordan. It also assessed the influence of work alienation and sociodemographic factors on readiness for professional development and willingness to learn. We used a cross-sectional correlation study design using the Arabic readiness for professional development and willingness to learn and work alienation scales among 328 nurses working in Jordan University Hospital, Amman-Jordan. Data were collected during the period of October and November 2021. Data were analyzed using descriptive statistics (Mean ± Standard deviation), Pearson correlation coefficient ($r$), and regression analysis. The perceived levels of work alienation (3.12 ± 1.01) and readiness for professional development and willingness to learn (3.51 ± 0.43) among nurses were found to be at high levels during this era. Work alienation was negatively associated with readiness for professional development and willingness to learn ($r$ = -0.54, $p$ <0.001). The higher educational level of a nurse was found to be associated with a higher work alienation ($r$ = -0.16, $p$ = 0.008). Results indicated that work alienation had a direct influence on readiness for professional development and willingness to learn among nurses ($R^2$ = 0.287, $p$ < 0.001). Work alienation among nurses seems to be increased during the pandemic and it has reduced their readiness for professional development and willingness to learn. Nurse managers at hospitals must assess the perceived level of work alienation among nurses annually and design appropriate counseling interventions programs to reduce their work alienation and increase their willingness to learn.

**Data Availability Statement:** The data underlying the results presented in the study are available on

Figshare (https://doi.org/10.6084/m9.figshare.19425572.v1).

**Funding:** The author(s) received no specific funding for this work.

**Competing interests:** The authors have declared that no competing interests exist.

## Introduction

Work alienation has a negative impact on nursing profession and may affect professional nursing development and willingness to learn during the era of coronavirus disease 2019 (COVID-19). This pandemic caused several psychological problems for healthcare providers (HCPs) by leading to extra workload, physiological deteriorations, and lack of willingness to learn and professional development [1]. There is evidence that nurses during COVID-19 have reported higher levels of missed care [2], extra work hours [3], and higher burnout levels [4]. These challenges may influence their readiness for professional development and willingness to learn.

Professional development can be defined as the lifelong process in learning activities that develop/maintain an individual's competencies, strengthen professional practice, and support their career goals [5]. Professional development among nursing professions is not only needed to keep abreast of patient care [5], but also affect self-motivation, support organization, and reinforce workplace [6, 7]. Willingness to learn is a significant indicator to learn about others' success or failure, which can be regarded as the antecedent or precursor of subsequent learning activities [8]. It is a solid guarantee to maintain the learning process. However, the barriers of willingness to learn among nurses refer to several issues manifested in the process of learning such as payment issues [9], leadership issues [10], and time constraints [11]. Readiness for professional development and willingness to learn have become an important aspect among nurses to acquire sufficient knowledge and skills, stick to standards of global care, and improve their professional status [12] that needs urgent attention from decision-makers. Work alienation in the workplace can be one of the major challenges influenced nurses to professional development and willingness to learn that has been very few studies to date on this issue.

The work alienation of nurses due to loneliness [13], as well as workplace trauma exposure [14] is a critical element at nurses' workplace. Thus, adverse experiences of this feeling in nurses were positively associated with missed nursing care [15] and were negatively associated with job performance and organizational commitment [13, 16]. For one thing, previous studies suggested that there were positive associations between work alienation, job burnout, and turnover intention [17, 18]. For another thing, a previous study revealed that the enrichment of work-to-family was a significant link between work alienation and work effort [16]. To our knowledge, relatively few studies have examined the association between sociodemographic factors and nurse work alienation.

Work alienation has five dimensions namely: meaninglessness, powerlessness, isolation, self-estrangement, and normlessness [19]. A few studies examined the work alienation during COVID-19 among nurses [20, 21]. The influence of work alienation on readiness for professional development and willingness to learn during the period of COVID-19 has not been investigated to date. However, several previous studies have investigated the impact of COVID-19 not only on physical status, but also on the psychological health of individuals [22, 23]. Based on self-determination theory, these psychological problems increase individual alienation [24].

## Materials and methods

### Aim

The aim of this study was to explore the perceived levels of professional development, willingness to learn, and work alienation during this pandemic among nurses in Jordan. It also assessed the influences of work alienation and sociodemographic factors including gender, marital status, age, site of work, and educational levels on readiness for professional

development and willingness to learn This study would provide insight into the influence of work alienation on readiness for professional development and willingness to learn during the COVID-19 pandemic.

## Design

A cross-sectional correlation study was used to explore the influences of work alienation on readiness for professional development and willingness to learn by distributing self-reported surveys conveniently to nurses during the COVID-19 pandemic. The STROBE checklist guidelines were followed. Please see S1 Checklist.

## Participants

Nurses in a tertiary hospital in Amman-Jordan were selected. The Jordan University Hospital consists of 34 departments divided into six sectors: operation/recovery rooms, intensive care units, medical/surgical floors, obstetrics and gynecology sectors, outpatient clinics, and educational center. Inclusion criteria were nurses at intensive care units, medical/surgical floors, and obstetrics and gynecology sectors. Besides, nurses who were working in different type of shifts, different educational levels, ages, experiences, and willing to participate were included in this study. Exclusion criteria were all nurses in operation/recovery rooms due to their nature of work. Moreover, nurses in education center were also excluded because they were supposed to have readiness for professional development and willingness to learn.

## Data collection

A convenience sample procedure was selected to recruit study participants. Approximately, the total number of nurses in the selected sectors was 500. We went to each supervisors' department in the selected sectors, and we asked them to distribute the self-reported surveys voluntarily to their nurses. As a result, a total of 328 nurses agreed to participate with a response rate of 65.6%. The self-reported surveys consisted of three parts: consent form, socio-demographic factors, and selected scales. Data was collected between October and November 2021.

## Study tool

**Sociodemographic factors.** We included sociodemographic factors of gender, marital status, age, site of work, and educational levels as independent variables.

**Readiness for professional development and willingness to learn scale.** This scale was used to assess the perceived level of readiness for professional development and willingness to learn among nurses during this era. This scale consists of 48-items divided into 2 dimensions; readiness for professional development and willingness to learn [12]. The first dimension consists of 3 subscales measuring the general development and learning as follows: (1) openness to changes in environment (12-items), (2) professional mobility awareness (4-items), and (3) self-evaluation of past educational development (7-items). The second dimension also consists of 3 subscales to determine professional readiness for learning and development as follows: (1) community of educational and professional goals (9-items), (2) professional information demand (3-items), and (3) effectiveness of in-service training (4-items). The remaining 9 items were excluded due to their role in buffer function. Before being used in Arabic language, we translated it from English language into Arabic language by following translation process and validation process.

*Translation process.* Translation and back-translation was completed by English-Arabic experts. Furthermore, a total of 4 expert panels specialized in nursing and education was agreed on a final version of Arabic readiness for professional development and willingness to learn scale with a percentage of 85%.

*Validation process.* We conducted face, content, and construct validity as follows: We conducted a preliminary study among 10 registered nurses to express their opinions on the Arabic version of readiness for professional development and willingness to learn scale. It was measured on a 3-point Likert type scale (appropriate/not appropriate, suitable/not suitable, and clear/not clear). On each item, an importance score of 1.5 or more would be sufficient to accomplish face validity: [25]. In this regard, all items on the Arabic version of readiness for professional development and willingness to learn scored above 1.5. A pilot study was also conducted with 8 experts (PhD holders) in nursing, psychology, education, and counseling to set their comments on the Arabic version of readiness for professional development and willingness to learn scale items. A percentage of 0.90 was scored from the opinion of experts and based on Lawshe's Table for content validity ratio, a percentage of 0.75 was deemed sufficient on 8 experts [26]. Finally, exploratory factor analysis by maximum likelihood method was used to calculate factor analysis. Factor loading (varimax rotation) ≥ 40, eigenvalue ≥1, Kaiser-Meyer-Olkin (KMO) test > 0.60, and Bartlett's test of sphericity ($p < 0.05$) were calculated [27] and Table 1 shows the result.

All items in the Arabic version of readiness for professional development and willingness to learn items scored above 0.40. Six factors were loaded with eigenvalues more than 1. The total variation of readiness for professional development and willingness to learn was 52.88%. The KMO test was 0.89. Bartlett's test of sphericity was significant (Chi-square ($\chi$2) = 7869.99, $p < 0.001$). Internal consistency was measured by calculating Cronbach alphas' value for the Arabic version of readiness for professional development and willingness to learn scale and its dimension. The overall Cronbach value was 0.88, and for dimensions it was 0.85 and 0.81 for readiness for professional development and willingness to learn, respectively.

**Work alienation scale.** This scale was used to measure work isolation, separation, powerlessness, and meaninglessness in hospitals. This study used previously published paper [28]. The selected scale was previously used in different Arab countries, and it has good psychometric properties [29, 30]. Cronbach's alpha was 0.93.

Readiness for professional development and willingness to learn and work alienation scales were measured on a five-point Likert type scale from 5 "strongly agree" to 1 "strongly disagree". for the negative items in both scales. We switched the Likert scale from 1 "Strongly Agree" to 5 "Strongly Disagree." for the negative items. The higher average score in two scales reflects higher perception levels of readiness for professional development, willingness to learn, and higher work alienation. To illustrate the cutoff point, we used the following equation based on average score = (Upper score—Lower score)/levels (mild, high). Thus, the average score lower or equal to 3 indicated a low level and average higher than 3 indicated a high level.

## Ethical considerations

The study was approved by the institutional review board at Jordan university hospital (No: 10/2021/17848). Informed consent was obtained from each participant by signing a consent form. We followed the declaration of Helsinki reporting guidelines such as; anonymous personal information and voluntary participation. We added corresponding author email regarding any questions from participants.

**Table 1. Factor loading for the Arabic version of readiness for professional development and willingness to learn scale (n = 328).**

| Items | Factor 1 OP | Factor 2 PM | Factor 3 SE | Factor 4 CE | Factor 5 PI | Factor 6 EI |
|---|---|---|---|---|---|---|
| Q1 | 0.72 | | | | | |
| Q2 | 0.69 | | | | | |
| Q3 | 0.66 | | | | | |
| Q4 | 0.65 | | | | | |
| Q5 | 0.65 | | | | | |
| Q6 | 0.66 | | | | | |
| Q7 | 0.83 | | | | | |
| Q8 | 0.77 | | | | | |
| Q9 | 0.79 | | | | | |
| Q10 | 0.71 | | | | | |
| Q11 | 0.74 | | | | | |
| Q12 | 0.61 | | | | | |
| Q1 | | 0.82 | | | | |
| Q12 | | 0.79 | | | | |
| Q3 | | 0.85 | | | | |
| Q4 | | 0.86 | | | | |
| Q1 | | | 0.67 | | | |
| Q2 | | | 0.65 | | | |
| Q3 | | | 0.72 | | | |
| Q4 | | | 0.65 | | | |
| Q5 | | | 0.77 | | | |
| Q6 | | | 0.59 | | | |
| Q7 | | | 0.82 | | | |
| Q1 | | | | 0.98 | | |
| Q2 | | | | 0.77 | | |
| Q3 | | | | 0.55 | | |
| Q4 | | | | 0.81 | | |
| Q5 | | | | 0.86 | | |
| Q6 | | | | 0.79 | | |
| Q7 | | | | 0.45 | | |
| Q8 | | | | 0.84 | | |
| Q9 | | | | 0.74 | | |
| Q1 | | | | | 0.81 | |
| Q2 | | | | | 0.91 | |
| Q3 | | | | | 0.69 | |
| Q1 | | | | | | 0.82 |
| Q2 | | | | | | 0.85 |
| Q3 | | | | | | 0.63 |
| Q4 | | | | | | 0.99 |
| Initial eigenvalues | 7.24 | 1.91 | 1.65 | 5.03 | 2.16 | 1.27 |
| Percentages of variance explained | 31.48 | 8.32 | 7.21 | 31.46 | 13.50 | 7.93 |
| Cumulative % | 31.48 | 39.80 | 47.01 | 31.46 | 44.95 | 52.88 |

Q: question. OP: Openness to changes in environment. PM: Professional mobility awareness. SE: Self-evaluation of past educational development. CE: Community of educational and professional goals. PI: Professional information demand. EI: Effectiveness of in-service training.

## Data analysis

Quantitative data was analyzed by using the statistical package for social sciences (SPSS V. 22). Descriptive statistics were used to explore the perceived level of readiness for professional development, willingness to learn, and work alienation. Regression analysis (stepwise method), Pearson correlation coefficient ($r$), t-test, and one-way analysis of variance (ANOVA) were used to assess the influences, associations, and differences between the main variables. The $p$-value was considered at 0.05.

## Results

There were 328 participants in this study. The majority of participants were male, married, and had bachelor degrees. Half of the participants were between 30 and 39 years old. Participants from intensive care unit account for 46.6% of the three recruited sectors in this study (Table 2).

Educational level factor was negatively associated with work alienation ($r$ = -0.16, F = 4.94, $p$ = 0.008). Based on average scores, the postgraduate educational level of a nurse the more prone s/he is to work alienation. The total readiness for professional development and willingness to learn was negatively associated with work alienation ($r$ = -0.54, F = 3.07, $p$ <0.001). Other sociodemographic factors were not associated with work alienation (Table 2).

**Table 2. Demographic factors and their association with work alienation among nurses (n = 328).**

| Outcome | Variable | Frequency (%) | M | SD | r | 95% CI | | t/F-distribution | p-value |
|---|---|---|---|---|---|---|---|---|---|
| | | | | | | Upper | Lower | | |
| **Work alienation** | **Gender** | | | | | | | | |
| | Male | 206 (62.8) | 3.15 | 0.96 | -0.02 | 0.27 | -0.19 | 0.36 | 0.72 |
| | Female | 122 (37.2) | 3.10 | 1.04 | | 0.26 | -0.18 | | |
| | **Marital status** | | | | | | | | |
| | Single | 55 (16.8) | 3.29 | 1.00 | -0.05 | 3.56 | 3.02 | | |
| | Married | 261 (79.6) | 3.08 | 1.02 | | 3.20 | 2.96 | 1.04 | 0.35 |
| | Divorce/widow | 12 (3.7) | 3.24 | 0.94 | | 3.84 | 2.64 | | |
| | **Age (years)** | | | | | | | | |
| | 20–29 | 78 (23.8) | 3.09 | 1.01 | 0.01 | 3.34 | 2.84 | | |
| | 30–39 | 186 (51.2) | 3.14 | 0.99 | | 3.29 | 2.99 | 0.09 | 0.91 |
| | ≥40 | 82 (25) | 3.11 | 0.97 | | 3.32 | 2.89 | | |
| | **Site of work** | | | | | | | | |
| | Intensive care units | 153 (46.6) | 3.16 | 0.65 | | 3.31 | 3.01 | | |
| | Medical/surgical floor | 110 (33.5) | 3.11 | 1.05 | -0.02 | 3.30 | 2.91 | 0.34 | 0.70 |
| | Obstetrics and Gynecology | 65 (19.9) | 3.04 | 1.09 | | 3.31 | 2.77 | | |
| | **Educational level** | | | | | | | | |
| | Diploma degree | 47 (14.3) | 2.71 | 1.13 | -0.16 | 3.04 | 2.38 | 4.94 | 0.008** |
| | Bachelor's degree | 241 (73.5) | 3.17 | 0.95 | | 3.29 | 3.05 | | |
| | Postgraduate | 40 (12.2) | 3.30 | 1.16 | | 3.67 | 2.93 | | |
| | Total Readiness for professional development and willingness to learn | - | 3.51 | 0.43 | -0.54 | 3.46 | 3.56 | 3.07 | 0.001*** |

M: Mean. SD: Standard deviation. $r$: Pearson Correlation coefficient. CI: Confidence interval. t: t-test.

**$p$-value < 0.01.

***$p$-value < 0.001

**Table 3. Descriptive statistics and overall levels of the study variables (n = 328).**

| Variable | | M | SD | Overall levels |
|---|---|---|---|---|
| Work alienation | | 3.12 | 1.01 | High |
| Dimension 1 Readiness for professional development | 1-Openness to changes in environment | 3.93 | 0.57 | High |
| | 2-Professional mobility awareness | 2.99 | 0.56 | Low |
| | 3-Self-evaluation of past educational development | 3.83 | 0.49 | High |
| | Total level of dimension 1 | 3.74 | 0.43 | High |
| Dimension 2 Willingness to learn. | 1-Community of educational and professional goals | 3.18 | 0.81 | High |
| | 2-Professional information demand | 3.56 | 0.59 | High |
| | 3-Effectiveness of in-service training | 2.89 | 0.73 | Low |
| | Total level of dimension 2 | 3.18 | 0.60 | High |
| Total level of Readiness for professional development and willingness to learn | | 3.51 | 0.43 | High |

M: Mean. SD: Standard deviation.

## Perceived level of the selected variables

The perceived levels of readiness for professional development, willingness to learn dimensions, and work alienation are presented in Table 3.

During the period of pandemic, work alienation exhibited a high perceived level. Professional mobility awareness and effectiveness of in-service training subscales demonstrated low level. Total level of readiness for professional development and willingness to learn were found to be at high level (Table 3).

## Regression analysis with work alienation

Work alienation, gender, marital status, age, site of work, and educational levels were studied with readiness for professional development and willingness to learn and the result of linear regression analysis is presented in Table 4.

Table 4 shows that work alienation had a direct influence on readiness for professional development and willingness to learn ($R^2 = 0.287$, $p < 0.001$). Multicollinearity was repudiated. Sociodemographic factors had no influence on readiness for professional development and willingness to learn.

## Discussion

With the continues evaluation of COVID-19 pandemic and its negative consequences toward nurses, the lack of readiness for professional development and willingness to learn caused by

**Table 4. Regression analysis by stepwise method.**

| Model | R | R Square | R Square change | Std. Error of the Estimate | Unstandardized Coefficients | | t | Sig. | VIF |
|---|---|---|---|---|---|---|---|---|---|
| | | | | | B | Std. Error | | | |
| 1[a] | 0.536 | 0.287 | 0.287 | 0.366 | 4.225 | 0.066 | 64.264 | <0.001*** | 1.00 |
| | | | | | -0.230 | 0.020 | | | |

VIF: Variance inflation factor. Dependent Variable: readiness for professional development and willingness to learn. t: t-test.

***$p$-value <0.001

[a] Predictors: (constant), work alienation

work alienation in hospital is ascendant. The findings of current study showed that the participating nurses perceived high levels of work alienation, readiness for professional development, and willingness to learn. Readiness for professional development and willingness to learn and educational levels were negatively associated with work alienation. Additionally, work alienation directly influenced readiness for professional development and willingness to learn among nurses during COVID-19.

A study conducted by [31] to investigate the work alienation among 62 university professors, they found that work alienation increased during pandemic when measured over three time points, this highlights the negative effects of work alienation in different professions. Nurses during COVID-19 often work extra time, feel separated from their families, and lack financial rewards, which can lead to accelerated work alienation among them. Another study was conducted among 306 nurses working in private healthcare providers in Oman, and their finding showed that the level of work alienation was to be at the moderate level [20]. This is inconsistent with our study findings. A study conducted in Egypt to explore the perceived work alienation among nurses during COVID-19 pandemic in intensive care units and inpatients wards using the same instruments. Their participants perceived high levels of work alienation. This could be rationalized by similarities in culture and the healthcare demands in which Jordan and Egypt have high level of demand [21]. Therefore, future studies are recommended to measure work alienation in other Arab countries.

In the current study, work alienation is associated with educational levels. As for the factor of educational levels, the higher educational levels occupied a higher proportion in work alienation. This could be rationalized by nurses who had master or PhD degrees were more likely to have anxiety, stress, unsatisfied with their life situations. Our findings are consistent with [21].

Interestingly enough, the participants in the current study perceived their readiness to professional development and willingness to learn to be at high levels. Frontline nurses indicated high willingness to learn which raised their professional commitment during COVID-19 pandemic [32]. Despite the high perceived levels of work alienation, participating nurses exhibited higher proportions of readiness for professional development and willingness to learn, which highlights the importance of readiness for professional development and willingness to learn during such crisis. Another study found that there were factors affecting nurses' intention to care for patient with COVID-19 such as age, department, clinical skills, and experience which could be directed to professional development and willingness to learn [33]. No studies explore these variables among nurses during COVID-19 pandemic till now. Future research is required to have a solid knowledge and reduce the scientific gap in this field.

Our findings reveal that work alienation directly influences readiness for professional development and willingness to learn. Work alienation has devastating effects on nurses' motivation and ability to learn. Another study found that work alienation is a mediating effect on the professional skills development [34]. A detrimental increase in work alienation has been noted among participating nurses, including powerlessness, meaninglessness, and isolation. Our findings highlight that work alienation can skew quality of patients care. Up to our knowledge this is the first study highlights effect of work alienation on willingness to learning and readiness for professional development among nurses during the COVID-19 pandemic which could be considered the main strength of the current study. In addition to, we did psychometric properties for the Arabic version of readiness for professional development and willingness to learn scale and the findings confirmed its validity and reliability. However, our findings are limited by using self-reported questionnaires, using a convenience sampling method, and using a single setting. Furthermore, we did not explore the perceived level of work alienation

among nurses in operation/recovery rooms and education center. Future research is recommended by using different settings.

## Conclusion

We conclude that work alienation influences nurses' professional development and willingness to learn which could reduce the quality of patient care during such pandemic. Nurse managers at hospitals should assess the perceived level of work alienation among nurses annually and design appropriate counseling interventions programs to reduce their work alienation and increase their willingness to learn. Additionally, providing continuous education is highly required by stakeholders and policy makers in order to increase nurses' satisfaction level and deliver better patient care.

## Supporting information

**S1 Checklist. The RECORD statement—Checklist of items, extended from the STROBE statement, that should be reported in observational studies using routinely collected health data.**
(DOCX)

## Acknowledgments

The authors would like to thank all nurses for their valuable contribution in this study.

## Author Contributions

**Conceptualization:** Othman A. Alfuqaha, Ohood F. Shunnar.

**Data curation:** Othman A. Alfuqaha, Ohood F. Shunnar.

**Formal analysis:** Othman A. Alfuqaha, Reema A. Khalil.

**Funding acquisition:** Tareq Z. Al Amad.

**Investigation:** Othman A. Alfuqaha, Fadwa N. Alhalaiqa.

**Methodology:** Othman A. Alfuqaha, Ohood F. Shunnar, Fadwa N. Alhalaiqa, Yazan Al Thaher.

**Resources:** Reema A. Khalil, Yazan Al Thaher, Uday M. Al-masarwah, Tareq Z. Al Amad.

**Software:** Ohood F. Shunnar, Reema A. Khalil, Uday M. Al-masarwah.

**Supervision:** Othman A. Alfuqaha, Fadwa N. Alhalaiqa.

**Validation:** Othman A. Alfuqaha, Fadwa N. Alhalaiqa.

**Visualization:** Othman A. Alfuqaha.

**Writing – original draft:** Othman A. Alfuqaha, Ohood F. Shunnar, Reema A. Khalil, Fadwa N. Alhalaiqa.

**Writing – review & editing:** Othman A. Alfuqaha, Ohood F. Shunnar, Fadwa N. Alhalaiqa, Yazan Al Thaher, Uday M. Al-masarwah, Tareq Z. Al Amad.

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
