## [Decision Letter · Decision Letter 0]

9 Mar 2023

PONE-D-22-27877Work Alienation Influences Nurses’ Readiness for Professional Development and Willingness to Learn: A Cross-Sectional Correlation StudyPLOS ONE

Dear Dr. Alfuqaha,

Thank you for submitting your manuscript to PLOS ONE. After careful consideration, we feel that it has merit but does not fully meet PLOS ONE’s publication criteria as it currently stands. Therefore, we invite you to submit a revised version of the manuscript that addresses the points raised during the review process.

We look forward to receiving your revised manuscript.

Kind regards,

Fatma Ay, Ph.D

Academic Editor

PLOS ONE

Journal Requirements:

Additional Editor Comments (if provided):

Dear Authors, congratulations for the research. We kindly ask you to send the article back to the system by making arrangements according to the referee's suggestions in the appendix.

Reviewers' comments:

Reviewer's Responses to Questions

**Comments to the Author**

1. Is the manuscript technically sound, and do the data support the conclusions?

Reviewer #1: Yes

2. Has the statistical analysis been performed appropriately and rigorously? 

Reviewer #1: I Don't Know

3. Have the authors made all data underlying the findings in their manuscript fully available?

Reviewer #1: No

4. Is the manuscript presented in an intelligible fashion and written in standard English?

Reviewer #1: Yes

5. Review Comments to the Author

Reviewer #1: The manuscript is well written and the topic is very interesting because it reflects the work alienation during COVID-19 amongst nurses, and its influence on nurses' readiness for professional development and willingness to learn.

I did not assess the statistical analysis rigorously because of my limited strength in quantitative studies.

The authors should add the key figures of the results in the abstract section.

The authors should use STOBE checklist to check the completeness of their methods section.

The authors used a convenience sample. They should add a discussion on the limitation of using the convenience sample.

6. PLOS authors have the option to publish the peer review history of their article (what does this mean?). If published, this will include your full peer review and any attached files.

Reviewer #1: No

---

## [Author Response · Author response to Decision Letter 0]

13 Mar 2023

Response to Reviewers

Dear editor,

We would like to thank you and the reviewer for their valuable comments and feedbacks. Point-by-point responses to reviewers are listed below.

Reviewer #1: 

Comment 1:

The manuscript is well written and the topic is very interesting because it reflects the work alienation during COVID-19 amongst nurses, and its influence on nurses' readiness for professional development and willingness to learn.

Response 1:

Thank you for your valuable comments.

Comment 2:

I did not assess the statistical analysis rigorously because of my limited strength in quantitative studies.

Response 2:

Thank you for your valuable comments.

Comment 3:

The authors should add the key figures of the results in the abstract section.

Response 3:

As your suggestion, we added key figures to our abstract section. 

The authors should use STOBE checklist to check the completeness of their methods section.

Comment 4:

As your suggestion, we added an appendix of STROBE checklist to our revised manuscript.

The authors used a convenience sample. They should add a discussion on the limitation of using the convenience sample.

Response 4:

In response to this comment, we added the convenience sample method in our limitation section. Please see our revised manuscript.

We hope now that our revised manuscript is acceptable for publication.

---

## [Editor Report · Decision Letter 1]

13 Apr 2023

Work Alienation Influences Nurses’ Readiness for Professional Development and Willingness to Learn: A Cross-Sectional Correlation Study

PONE-D-22-27877R1

Dear Dr. Alfuqaha,

We’re pleased to inform you that your manuscript has been judged scientifically suitable for publication and will be formally accepted for publication once it meets all outstanding technical requirements.

Kind regards,

Fatma Ay, Ph.D

Academic Editor

PLOS ONE
---

## [Editor Report · Acceptance letter]

25 Apr 2023

PONE-D-22-27877R1 

Work Alienation Influences Nurses’ Readiness for Professional Development and Willingness to Learn: A Cross-Sectional Correlation Study 

Dear Dr. Alfuqaha:

I'm pleased to inform you that your manuscript has been deemed suitable for publication in PLOS ONE. Congratulations! Your manuscript is now with our production department. 

Kind regards, 

on behalf of

Dr. Fatma Ay 

Academic Editor

PLOS ONE